# Unintended Pedagogical Consequences of Emergency Remote Teaching at a Rural-Based University in South Africa

**Siyabonga Theophillus Pika [1,*] and Sarasvathie Reddy [2]**

1. Department of Accounting, Walter Sisulu University, Butterworth 4960, South Africa
2. School of Education, University of KwaZulu-Natal, Durban 3605, South Africa
* Correspondence: sphika@wsu.ac.za

**Abstract:** In this empirical article, we argue that while emergency remote teaching (ERT) may have achieved its goal of saving the academic years during the COVID-19 pandemic, it also constructed unintended pedagogical consequences that were possibly overlooked at the time of advocating for it. We also contend that students and lecturers from rural-based universities (RBUs) in South Africa experienced different unintended pedagogical consequences compared to their counterparts who belong to urban-based universities (UBUs). Thus, the research question that the article raises is as follows: What were the unintended pedagogical consequences that students and lecturers based at RBUs experienced during the transition to ERT? Drawing on students' and lecturers' lived experiences of ERT, this article foregrounds unintended pedagogical consequences that arose at one RBU in South Africa during the transition from face-to-face teaching to ERT. Underpinned by the tenets of critical realism philosophy, as well as student integration theory, in-depth interviews with three lecturers and six students were conducted. The findings of the study indicate that home conditions, individual characteristics, pre-COVID-19 blended learning experiences, university training and support, teaching, learning, assessment practices, and policies altogether contributed to the construction of unintended pedagogical consequences of ERT presented in this article. These consequences include (1) the exclusion of low-income students from active teaching and learning, (2) equipping middle-class students with better chances of success than working-class students, (3) distressing female students and lecturers more than their male counterparts, and (4) unproductive assessment practices. This study may be beneficial to academics and policymakers from similar contexts in their plight to continue with remote teaching and assessment (RTA) after the pandemic.

**Keywords:** COVID-19 lockdown; critical realism; emergency remote teaching; higher education; rural-based university; unintended pedagogical consequences

## 1. Introduction

South Africa is viewed as the most unequal country in the world, and this inequality is a large determinant of the country's high poverty rate [1]. The social inequalities in South Africa can be best understood by studying the apartheid policies [2] that produced them. The apartheid policies implemented in South Africa between 1948 and 1994 promoted white supremacy by fostering a culture of discrimination against the majority of non-white South Africans [3]. The Bantu Education Act (No. 47 of 1953) classified and separated education along racial lines (White, Indian, Coloured, and African education departments) [2] and the Extension of Universities Act (No. 45. of 1959) established higher education institutions for "non-whites" which were placed in deep rural areas [3]. This resulted in the establishment of separate universities for white and non-white students who attended urban-based universities (UBUs) and rural-based universities (RBUs), respectively. The realization of such policies rendered non-white students with inferior education and fewer learning opportunities as opposed to the white students who attended UBUs [2,3]. Subsequently, since the dawn of democracy in 1994, South African higher education revisited its policies

to address the past inequalities that were caused by the apartheid regime. The three resultant types of universities in South Africa are traditional universities, universities of technology, and comprehensive universities [4] that were established with the primary aim of expanding access for marginalized Black South Africans. This article describes a study conducted at a comprehensive RBU that was established in the year 2005 through the merger of two former Technikons with a traditional university [5]. Due to structural and cultural differences between the merged institutions that were all historically disadvantaged, it was difficult to establish a cohesive merged university that would be well-resourced [2,3].

Despite the modification of policies, numerous systematic segregation practices still exist, and because of their racial, ethnic, socioeconomic, or geographic backgrounds, many students are unable to participate fully in their own learning or make use of current educational resources. Some scholars believe that international standards are already attempting to impose a colonial education model in South Africa [2]. From the early 1990s onwards, many Black students who had performed well in their school-leaving examinations preferred to enroll in the better-resourced historically white UBUs that were now available to them [4]. While all universities recruit students from both urban and rural communities, rural middle-class students with better matric grades typically enroll in UBUs because they have better teaching and learning resources [4], even today. In the end, middle-class students are largely absent at historically Black RBUs [4]. Irrespective of their social class, students with higher matriculation grades have a greater chance of being awarded grants and scholarships. In contrast, students who perform poorly in their matriculation exams are typically the ones who enroll in RBUs. Numerous RBU students fall into this category, and some academics refer to them as underprepared students for university [4]. In contrast to their urban counterparts, RBUs are underfunded and under-resourced, making it difficult to recruit and retain highly experienced professors to teach in rural contexts.

This situation is not exclusive to South Africa. Research around the world suggests that young people who have the most access to and success in higher education are the children of middle-class, educated caregivers [4]. Since school-leaving examination performance and conditions in the home of origin are associated with the ability to access better schools, social class [4] is an increasingly important indicator in enrolment patterns across the globe [2,4]. This historical context of RBUs is provided to justify the argument that because of apartheid policies, UBUs have earned substantial advantages over their rural counterparts, and the inequities between them are extremely large in many respects, including human resources, teaching and learning facilities, the academic performance of students, financial status, research capacity, and the digital divide [6] among staff and students.

The transition from face-to-face pedagogy to technology-based ERT because of the COVID-19 pandemic occurred when the imbalances [7] described above persist between UBUs and RBUs. Long before the pandemic, RBUs grappled with inadequate teaching and learning facilities [2]. Subsequently, students' access to learning resources and academic support was limited during the transition to ERT at RBUs. Furthermore, some lecturers teaching at RBUs lacked the technological [5,8,9] and pedagogical expertise required to teach online and/or in blended learning environments [10]. Although ERT was deemed to be the most viable pedagogical solution during the time of the pandemic, its implementation was unplanned [5] and may not have been appropriate for everyone at the RBUs. In this article, we argue that, while ERT may have achieved its goal of saving the academic years during the COVID-19 pandemic, it also highlighted unintended pedagogical consequences that were possibly overlooked at the time of advocating for it. Although the unintended consequences discussed in this article may be experienced elsewhere, we argue that the extent of their materialization differs from context to context. A qualitative understanding of students' and lecturers' experiences of the transition to ERT was, therefore, necessary to understand the unplanned pedagogical consequences that arose during the transition, especially in the context of RBUs.

## 2. Emergency Remote Teaching at the Researched University

Emergency remote teaching is well documented in the literature globally since the eruption of the coronavirus pandemic (see, for instance, [7–9]). Scholars have reported on the results of empirical studies conducted in different contexts including higher education. However, only a few studies were conducted at the researched site [5,8,9] and other RBUs in South Africa. Given their segregated nature, some universities were able to adapt more seamlessly to the remote teaching and learning environment than others [5]. The transition to ERT at the studied university was not simple. In response to national initiatives, the studied university adopted a primarily online, technology-infused instructional model with a distinct delivery strategy to replace the contact model [5]. Although a blended learning strategy was adopted prior to the pandemic, technology integration in teaching and learning was minimal [8,9]. Early in April 2020, a technical task force was established to develop online policies and other related guidelines. The team consisted of academic and non-academic personnel with knowledge of online and Information and Communication Technology (ICT) instruction. This team was instrumental in driving the online learning initiative at the researched university. The team, comprised of Deans, Campus Rectors, and other relevant personnel, met regularly to assess the implementation of agreed-upon interventions, and monitor the progress [5].

Subsequently, in line with the ERT strategy of the university, a pilot study was conducted at the research site and the resultant framework influenced the future direction of the university (see [5]). The special training programs on using the university's learning management system (LMS), Blackboard Learn, and videoconferencing tools such as Microsoft Teams and Zoom, were held to prepare lecturers to teach and assess students in remote settings. The training was conducted simultaneously with the distribution of laptops to students and lecturers. Lecturers collected their resources from the university, but students collected them at designated sites that were communicated with them. Considering that social gathering restrictions were in place, only a limited number of laptops could be distributed on a given day. Subsequently, the university transitioned to ERT later than its counterparts. Unlike other universities, most students at the studied university are funded by the National Student Financial Aid Scheme (NSFAS). Learning from the NSFAS policy implies that their parents' combined income is less than three hundred and fifty thousand rand (SAR 350 000.00) per annum [11]. However, given that such training took place during a time of high uncertainty, frustration, and anxiety [12], its impact may have been less positive than it otherwise would have been. Whether lecturers achieved the learning outcomes of the online training programs or not, they were still mandated to teach and assess students remotely adopting the underlying principle of accommodating every student. While recognizing the benefits of ERT for students, instructors, and the university community, the aim of this article is to foreground the unintended pedagogical consequences of ERT by drawing on the students' and lecturers' reported experiences of teaching, learning, and assessment as they engaged with ERT in the context of an RBU. As stated earlier, the research site is representative of a group of universities in South Africa with roots in the apartheid educational structures that deliberately limited the quality of educational opportunities available to Black social groups [4]. Most of this group's institutions are located outside of South Africa's major cities [13]. In the South African higher education literature, there is a dearth of studies conducted in these institutions [14] due to the apartheid past. Therefore, this study significantly contributes to this knowledge gap in the field of technology adoption by an RBU during the time of the pandemic.

As the pandemic spread, students were forced to leave university campuses [13] and return to their homes [15] of origin. The closure of university campuses had implications for teaching and learning [8,16,17], particularly in remote settings. A history of inadequate resourcing [14] and ongoing funding challenges [18] have resulted in difficulties in the provision and use of technology [19]. During the pandemic, the university (research site), with the assistance of the Department of Higher Education and Training (DHET), provided laptop computers and data to almost all its students. Furthermore, lecturers were trained to

manage tuition in an online or blended learning environment and to administer formative and summative assessments online. However, given that students were forced to study at home [14,15], and that many of them come from rural areas, with some rural areas [13] in the Eastern Cape lacking electricity [2], it is important to study the participants' experiences of ERT to understand the unintended pedagogical consequences that may have occurred because of the transition to ERT. The purpose of this article, therefore, was to highlight the unintended pedagogical consequences of transitioning to ERT that arose at an RBU during the lockdown periods of the pandemic.

### 3. Materials and Methods

This article employed a case study research approach and purposive sampling [12] to recruit the participants. One of the authors works as a lecturer at the research site and, therefore, access to the participants was easy. One of the authors made an open invitation to seventeen lecturers in a selected department. Three lecturers agreed to participate in the study. The participating lecturers were then requested to invite their students to participate in the study as well. Lecturers communicated with 150 students from their classes. More than ten students promised to participate in the study. We set different dates for lecturers' interviews and focus group discussions for more than ten students. Six students were available for the focus group discussion, but individual interviews were conducted because of the decline in the anticipated number. Ultimately, the sample size for this study comprised three female lecturers and six second-year students (three males and three females). The students joined the university in January 2020. The three lecturers taught the same course to three different groups of students. Two students (one male and one female) were purposively selected from each of the three groups. The course lecturers identified students from their groups whom they believed would express their views freely, regardless of their socioeconomic background. Interview schedules were prepared for students and lecturers. Interview questions were aligned with the constructs of the Students Integration Theory. During the two-month period of May and June 2021, in-depth interviews were conducted with students and lecturers. The interviews were semi-structured to understand participants' experiences with ERT. The interview schedules were prepared for both student and lecturer interviews. The interview questions were structured around the concepts of the student integration theory and prompted participants to reflect on their personal experiences of teaching, learning, and assessment in general, social life as students and academics, and their home conditions during the pandemic. Interviews were semi-structured, implying that follow-up questions were made during interviews and new unplanned questions emerged during interviews. No discipline-related questions were asked. Since the case study approach was used [12], and there is a possibility of annual variation in student enrollments and shifts in lecturers' pedagogical and technological experiences, trying to extrapolate results directly to other populations is neither reasonable nor valid. Generalization was not the purpose of the sample selection or the overall study. The concepts deconstructed from student integration theory [20] were used also to analyze participants' interview data and are understood in this study as the main structures and mechanisms that influenced the lived experiences of both students and lecturers (see Figure 1 below). Gatekeepers' permission was sought and granted for the study. Anonymization was applied to all data.

Drawing on critical realism philosophy, the study adopted a critical realist lens to identify the structures and mechanisms that influenced students' and lecturers' lived experiences of ERT. According to Bhaskar's critical realism, three layers of reality exist: the empirical domain, the actual domain, and the real domain [4]. The empirical domain captures participants' experiences and observations and the actual domain is the layer of events from which these observations and experiences emerge. The real domain captures structures and mechanisms that are understood to exist independently of human action and thought [16]. This contrasts with events in the actual domain and experiences and observations in the empirical domain, which are understood to be relative [4].

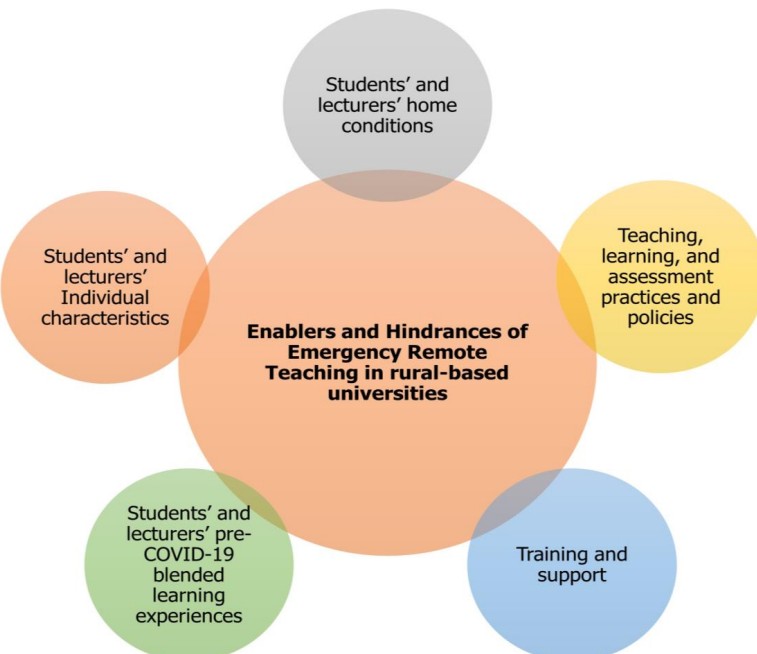

**Figure 1.** Enablers and hindrances of ERT at a rural-based university (Source: Authors).

Data were then subjected to a process of analysis involving abduction to identify the structures and mechanisms operating at the level of the real, in accordance with the tenets of critical realism [16] and student integration theory [20,21]. Abduction is the process of using theory to infer the existence of structures and mechanisms, as well as the interplay between them [4]. The concepts of student integration theory [20,21] were deconstructed as the explanatory theory in this abduction process, with the elements of the theory understood as structures and mechanisms located at the level of the real [4]. Critical realism acknowledges the existence of independent reality while also acknowledging the influence of human thoughts and actions on how we know and interpret that reality [4]. Critical realist researchers investigate the interaction of structures and mechanisms at the level of the real through the deductive process of abduction [4]. In moving from observations and experiences reported by participants to identify the enduring structures and mechanisms at the level of the real, critical realist researchers acknowledge their potential fallibility [4]. Any study based on critical realism must, therefore, check for fallibility using strategies such as member-checking and triangulation [4], and these processes were carried out during this study. The transcripts were sent back to the participants to verify the accuracy of the transcription conducted.

### 4. Results

The article sought to answer the following research question: What were the unintended pedagogical consequences that students and lecturers based at South African RBUs experienced during the transition to ERT? While many other concepts may exist, this study deconstructed concepts from student integration theory [20,21] to explain the findings. The results of this study are discussed according to the following five concepts, i.e., (1) Students' and lecturers' home conditions, (2) Students' and lecturers' individual characteristics, (3) Students' and lecturers' pre-COVID-19 blended learning experiences, (4) University training and support, and (5) Teaching, learning and assessment practices and policies. These concepts are understood as structures and mechanisms that triggered the emergence of participants' perspectives, (i.e., perceptions, practices, and experiences), from which the unintended consequences are drawn. It is important to note that these concepts are interlinked even though they were discussed separately. For instance, it is impractical to separate students' individual characteristics from their home conditions because of their interdependence. Figure 1 depicts these concepts.

*4.1. Students' and Lecturers' Home Conditions*

The interview data revealed that the student participants prioritized their choice of universities based on their family's affordability to pay for such university costs. Some student participants chose to attend RBUs on purpose because they could not afford tuition, housing, and other expenses at urban-based historically privileged universities. Some students enrolled at the studied university because it was closer to their homes. This was deemed necessary to save money on transportation to and from the university, as evidenced in the extract below from student participant 3:

> *I am the only child who passed grade 12 in my family. We all live here in the Eastern Cape at Ncise. I did not apply to other universities because they are far from home, and nobody is working at home. So, where would I get the money to travel when I wanted to see my child and my family? Accommodation is expensive. So, the NSFAS stipend would not be enough to provide for all my needs at other universities. At least now I can visit my family when I need to and support them financially with my NSFAS stipend when necessary.*

The extract above suggests that student participant 3 sometimes used her National Student Financial Aid Scheme (NSFAS) funds to support her family. This case may not have been unique to this participant, there could be many other students in a similar situation. This implies that NSFAS funds may, in some instances, be used to cover some unintended expenses, as student participant 3 has shown. The critical realist lens allows us to see that student participant 3's family background and her home's socioeconomic conditions influenced her decision to enroll in the studied university and to spend the funding in this manner.

In addition, some student participants indicated that they could not afford to buy extra data when the data provided by the university was depleted. Student participant 5 stated:

> *Data finishes before the month ends. Once that happens it becomes difficult to attend online classes. We can't even send emails or communicate with classmates on WhatsApp. It becomes worse when we must submit assignments or write online tests. We are forced to wait for the following month for the data to be reloaded*

Similarly, student participant 1 stated:

> *Sometimes we could not download notes, voice-over PowerPoint presentations, videos, and lecture recordings because data is not enough, it finishes before the month ends. We use the night data for downloads because it is more than the day data. But you can't use it to attend online lectures; I wish the university could increase the day data as well.*

To save data, students had to watch or download online videos or lecture recordings at night. This may have impacted their concentration levels during live lecture sessions during the day. Other challenges reported by student participants 2, 3, 4, and 6 included poor network connections, lack of electricity in some cases, and overcrowded households that made the environment detrimental to learning. Student participant 4 attested to the following statement:

> *There is no electricity at home, and we are many. Sometimes I helped my younger brothers with their homework because I am the only one with a computer at home. As a result, the battery and data do not last long. Even at res. (student residences) there is a lot of noise. Some students speak out loudly and some play loud music when they do not have classes. So, we do not hear properly sometimes during live online classes. Given a choice, I would prefer to attend face-to-face classes. Online classes are not good for me in many ways.*

The extract from student participant 4's interview data is the empirical evidence of the participant's reasons for his frustration with online classes and his preference for face-to-face classes. The critical realist lens allows us to see that students who come from low-income homes are likely to have experienced online teaching differently than students

who come from middle-class families. This finding indicates that laptops and data provided to university students to learn remotely may not have been sufficient to enable efficient ERT. More needs to be accomplished to extend battery life and network access for students who live in rural areas without electricity. The finding also proposes the need for the revision of students' allocation practices to residences. For instance, the students who registered for common qualifications and are accommodated together are likely to attend at the same time and work together fittingly. This arrangement could not only improve the efficiency of online classes but also improve the sense of belonging and related social aspects of students' life.

Student participants 1, 2, and 3 indicated that they spent much time doing household chores and ended up not having enough time for their studies. For instance, student participant 2 asserted:

*Studying from home was not easy for me. I had to use abnormal working hours to finish different activities. I had to strike the balance between domestic work activities and academic activities by waking up early and sleeping late at night. My typical day would start with making breakfast and cleaning, cooking during the day, and preparing supper. These were the activities I would not be doing if I were on campus. Sometimes I would be too tired that I could not complete the academic work in the way that I would if I were not at home.*

Female participants reported this constraint more than their male counterparts. Only one male student participant, participant 5, indicated having missed afternoon classes because of household chores. He asserted:

*Domestic work did not affect me that much. It was only Monday and Wednesday classes that were affected. They ended late at 16:30 pm and I had a responsibility of looking after cattle when I was at home. So, I had to leave at home around 16:00 pm more especially during Winter to look for cattle in the veld. Other than that, no other household chores affected my studies.*

The critical realist lens allows us to see that the social construction of gender roles by the rural communities where the student participants lived resulted in differing experiences of ERT among male and female participants.

Similarly, the home conditions of lecturer participants also contributed to the experiences that emerged in the adoption of ERT. One lecturer indicated that she has a study room that every family member respects. So, she makes time to prepare and record video lectures to share with students with ease. Lecturer participant 2 asserted,

*It really helped to make my husband and children understand and respect my privacy as a lecturer during ERT. For instance, I would tell my children not to disturb me once I was in the study room. I would then record my lecture videos peacefully. Even when I conducted live lecture sessions, my children would not disturb me. I don't know if I were to stay with my parents or in-laws at home; maybe I would be narrating a different story now. But my husband also respected my preparation and live lecture times.*

On the contrary, the other two lecturer participants, participants 1 and 3, reported having been struggling to secure a quiet space at home to record lecture videos and/or offer live lecture sessions. As a result, preferred to go to their offices to record videos or conduct online sessions. Lecturer participant 1 stated:

*The challenging part of ERT was that all my children were at home. I had to assist them in searching for information online to complete their assignments while I also had a task of preparing for my lectures. Balancing the responsibilities of being a mother and a lecturer was challenging. You could not run away from the household chores such as preparing food and cleaning, more especially when you have young children, you know! And hiring an assistant was risky at that time. The only viable solution was to use my office at work to record and conduct online sessions or use the quiet times at night to record videos while children were asleep.*

It could be observed from the finding presented above that lecturer participants' home conditions influenced the way they experienced ERT. The findings imply that the home conditions did not only influence the student participants, but they also exposed lecturer participants to similar challenges.

*4.2. Students' and Lecturers' Individual Characteristics*

When reviewing the set of transcripts of both the student and lecturer participants, a variation in the levels of technological skills and abilities was noted. Student participants attributed their level of skills and abilities directly to their basic education experiences. This may be evidenced by what student participant 6 shared: *"I was fortunate to be introduced to computer applications subject in my matric. The computer literacy skills that I had were improved as the result of online learning".* The critical realist analysis of this finding suggests that student participant 6 was likely to come from a middle-class home and attended one of the better-resourced schools that is likely to be a private school.

On the contrary, many student participants indicated that they had no prior experience using computers. For instance, student participant 1 stated:

> *It was very difficult for me to learn how to use a computer on my own without any previous experience. I had to spend much data watching YouTube videos on how to perform certain tasks using a computer and I was not good in searching for the relevant videos. I could not submit assignments on time because I was slow in typing and sometimes, I did not know how to perform certain tasks.*

The extract from the student participant above suggests that the perception of the adequacy of the data provided by the university to students could also be subject to the computer literacy skills of students. Computer-literate students could have spent the data differently; obviously not watching the same YouTube videos that the computer-illiterate student participants claimed to have watched. This finding confirms that students from low-income homes experienced ERT differently than students from middle-class homes.

Some students perceived online lectures as uninteresting compared to traditional face-to-face classes. They reported online teaching lacked debates, discussions, and demonstrations as learning strategies. Student participant 6, for instance, stated:

> *I found online teaching to be limiting the development of students' social skills. Some of us are talkative and understand the subjects better when we debate topics among ourselves as students. We need to improve our presentation skills because we need them in the workplace. For a lecture to be enjoyable, it needs to combine teaching methods that allow students to participate in learning; sometimes in teaching our peers and learn from one another. Online tests require us to answer multiple-choice questions most of the time. We are not given enough chance to explain our answers. This encourages us to memorize answers and I am not good in doing that. I prefer to express myself. But I do understand that some of us are not good at typing . . .*

The above extract suggests that online teaching may have been inadequate in engaging all students effectively in learning. The move to ERT seems to have supported students who preferred rote learning approaches and deterred students who adopted deep approaches to learning. Likewise, students who were computer-literate were better off than students who were computer-illiterate. Students providing similar extracts to the one presented above are likely to be students who had developed active learning skills in their prior schooling. Similarly, students who studied through rote learning in high school are likely to have enjoyed the assessment practices adopted in online tests unless they were stimulated otherwise.

Lecturer participants agreed that online summative tests were developed mainly using objective question types such as multiple-choice questions, true or false questions, fill-in-the-blank, and matching columns. Lecturer participant 1 asserted:

> *In ideal situations, a lecturer would want students to express themselves openly in online assessments by asking them open-ended questions. But given that some students*

*were computer illiterate, that would mean that most of them would not finish writing assessments on time. They would spend much of the time trying to type their answers, which might lead to anxiety and poor performance, not because they don't understand the subject content, but since they are not competent in the new assessment platform. So, I limited the number of open-ended questions I posed in summative assessments.*

The above extract suggests that the design of assessment tasks by some lecturers might have fallen short in assessing higher-order thinking and critical thinking skills, depending on the lecturers' perceptions of what it means to assess computer-illiterate students online and lecturers' competencies in formulating good assessment questions. This is another aspect that could be addressed through pedagogical training [15] of lecturers as assessors in online environments.

The general observation by both lecturer and student participants was that students' participation was restricted during online classes even if they were encouraged to speak. Lecturer participants attested that because of students' unwillingness to speak it was difficult to engage them meaningfully in class discussions. Student participant 3 stated: *"I could not speak during live lectures because I am a shy person"*. In contrast, student participant 5 stated that he participated better in live online lectures because he was shy. He said: *" . . . the fact that lecturers and classmates cannot see me when talking makes me confident to speak during online classes because I am a shy person"*. The language of instruction was reported as a barrier by many student participants. They were not confident in speaking English. Student participant 5, for instance, stated:

*I struggle to speak in public whether I speak face to face or in online environments. Ndiyathintitha (a phrase in IsiXhosa that means, I stutter). I become worse when I speak English. I can't speak English vocally; I prefer to write it. I am worried that I can make mistakes in my speech. I think about the class recording that will be shared with me having made the grammar mistakes. Yhoo! That does not sit well with me. So that is why I can't speak when the session is recorded. I don't want to embarrass myself.*

The above extract suggests that some students could have preferred to participate only when the virtual sessions were not recorded. Failure to record live sessions, though, disadvantaged students who could not attend live lectures because of network glitches and other reasons. This finding also suggests that some student participants could have deliberately excluded themselves in class discussions because they could not express themselves confidently in English, although they were encouraged to code-switch.

### 4.3. Students' and Lecturers' Pre-COVID-19 Blended Learning Experiences

The interviewed lecturer participants indicated that they did not use blended learning in their classrooms prior to COVID-19, except for one lecturer participant who indicated to have a fair knowledge of the learning management system and used it a few years before the pandemic. Lecturer participant 3 asserted:

*I started using Blackboard in 2015 after attending a training that the university organized. I used it mainly to share learning materials with students and to conduct formative and summative assessments. I conducted summative assessments in a controlled lab environment to avoid plagiarism. During the pandemic I didn't struggle much, instead, I was one of the e-learning champions who assisted in training colleagues in their departments to use the Blackboard for emergency remote teaching. I never used video conferencing software before the pandemic . . . I don't think the time for the training we had at the beginning of 2020 was enough. I was fortunate that I already started using blended learning way before the pandemic, but for someone who had no prior experience, I don't think they would have grasped all the ideas presented in the training in that short space of time.*

The extract from lecturer participant 3 confirms that lecturers' prior knowledge of LMSs influenced the ways they transitioned and adapted to ERT. Lecturers who had no prior knowledge of blended learning were likely to struggle to adapt to the new ERT environment.

Sharing the same sentiments, lecturer participant 1 asserted:

*Even though the training sessions prepared me to understand how Microsoft Teams, Blackboard, and Moodle work, I found it difficult to understand practical ways of involving students in discussions during live lectures. Also, I could not use enough discussion questions in assessments because most students were slow typists and could not finish writing timed assessments on time.*

The extracts from lecturer participants' interview data confirm that the training was not enough to prepare inexperienced lecturers to manage online classes effectively at the beginning of ERT. As a result, after the training, the common approach that some lecturers adopted was to upload eBooks, handouts, voice-over PowerPoint presentations, self-made videos, lecture recordings, and YouTube videos to the university's LMSs [15]. Students had to download the uploaded learning materials and read or listen to them offline to save data. Subsequently, students could not engage meaningfully with learning materials as they would in a traditional face-to-face class [15].

Student participant 4 attested:

*All university students were trained in using Blackboard and Moodle, but the time was not enough. I used Blackboard for the first time in 2020. When I started to understand it, the university shifted to Moodle. The time the university spent training us was not enough, but I managed to understand both apps by educating myself and watching YouTube videos.*

The extracts from both students and lecturer participants above suggest that the training that was provided at the beginning of 2020 that attempted to prepare both students and lecturers technologically to use the university's LMSs was not enough [15].

When prompted to comment on the underlying reasons for the preferred invisibility by students during live lectures, student participants mentioned saving data as the main reason for deactivating live videos. They also indicated that they were not comfortable subjecting their home conditions to the public. Student participant 5 reported:

*Lecturers share live lecture recordings with all students after the class. In most cases, the videos are not edited. This means that my home conditions may be exposed. As I am being recorded, whoever watches the recording will see me and the home environment during the time of the recording . . . There are certain things about my home condition that I would not like the public to see . . .*

The practice of maintaining anonymity in live lectures made it difficult for lecturers to see students who were listening attentively during lectures even though they were allowed to deactivate their videos. Sometimes students would sit in one place and share one computer to save data. This practice discouraged lecturers because they would think that few students had attended the lecture whereas there might have been more students attending than what the videoconferencing system showed. The opposite was true in some cases; lecturers would teach a few students thinking that those who had not logged on were sharing computers with friends. This implies that students' attendance was difficult to monitor and control during ERT because of the reasons stated above.

*4.4. Training and Support for Both Students and Lecturers*

The student participants acknowledged that the university had support structures in place to provide a smooth transition to ERT during the pandemic; however, they believe that it was not enough to support them both academically and socially during ERT, as student participant 6 attested:

*. . . Sometimes the phone numbers that we were given for academic support were not picked up and at other times as students, we did not have airtime to phone them. Where email addresses were given, there was a challenge of delayed responses. Maybe that could be because of the large number of students requesting the same services or because of the network challenges... Access to an online library was also difficult because it needs data,*

*network connection, and electricity. The location of my home in a rural area made it difficult to access learning materials from the library.*

Due to insufficient data and increased network challenges, some students struggled to collaborate with their peers and to communicate with their lecturers while they were studying from home. Subsequently, some students felt isolated and depressed and ended up deregistering from some courses that they believed were problematic. Student participant 1 attested to this claim by saying:

*I don't want to lie. I was tempted to cancel the registration of some of my modules as some of my friends did. I had no hope that I would manage to study so many modules independently because I am used to studying in groups with my friends. Thank God, I did not cancel them because I managed to pass all of them through the support that I received from the Writing Centre of the university and the WhatsApp support group that my classmates created.*

It could be seen from the analysis of participants' interview data that the availability of network connections and data was critical in all participants' lived experiences of ERT. Their availability correlated to better experiences of ERT while their unavailability related to worst experiences. The socioeconomic conditions of students' homes strongly emerged as structures at the level of the real that influenced students' home conditions and the availability of data.

### 4.5. Teaching, Learning, and Assessment Practices and Policies

As evidenced in the studied university's website, the university revised its teaching, learning, and assessment policies to accommodate ERT. When ERT was adopted, the policies encouraged the adoption of any educational technologies that could assist lecturers in their teaching endeavors. However, summative assessments were restricted to the university's approved learning management systems (LMS), Blackboard, and Moodle. Lecturer participant 3 stated:

*Blackboard was the LMS that the university used since 2009, but when the university shifted to ERT the version of Blackboard that the university used became overloaded and difficult to maintain, triggering the move to its cloud-based platform that became much more expensive. Subsequently, the university adopted a new LMS, Moodle, that was much cheaper than Blackboard. However, the shift to Moodle necessitated another training to equip both lecturers and students. Then again, the training provided was not enough to prepare lecturers to engage students meaningfully in learning and assessing higher-order thinking and critical thinking.*

The problem with online exams was that the integrity of assessments could not be verified. Students may have shared their login passwords with acquaintances who may have been asked to write on behalf of enrolled students, or students may have written individual exams in groups, according to lecturer participants. Respondus Lockdown Browser and Respondus Monitor were used as proctoring tools by the institution to prevent cheating during online assessments. Due to network issues and restricted bandwidth, the quality of Respondus Monitor clips was occasionally poor, making it difficult for lecturers to ascertain whether students had cheated or not. In such circumstances, lecturers have the discretion to allow students to repeat online examinations in a controlled setting in the lecturer's presence if they were suspected of cheating in the prior online assessment. Although the accuracy of Respondus in avoiding cheating cannot be guaranteed, it has been considered to assist in lowering students' probabilities of cheating, thereby contributing to enhancing the integrity of online exams.

## 5. Discussion

The findings of this study were explained in the previous section using the five concepts of students' integration theory. The concepts are (1) Home conditions of students and lecturers, (2) Individual characteristics of students and lecturers, (3) Pre-COVID-19 blended

learning experiences of students and lecturers, (4) University training and support, and (5) Teaching, learning, and assessment practices and policies. These concepts are interpreted as structures and mechanisms that triggered the emergence of participants' perceptions, practices, and experiences, from which the unintended pedagogical consequences of ERT are observed. The key findings that could be drawn from the results explained above are summarized as follows: (1) low-income students are excluded from active teaching and learning, (2) middle-class students have better chances of success than working-class students, (3) during ERT, female students and lecturers were more distressed than their male counterparts, and (4) unproductive assessment practices emerged during ERT. We will now discuss the findings of the study.

South Africa's multidimensional divide [3,6] that exists between UBUs and RBUs [19] may be reflective of a social connection prompted by apartheid's substantial political influence on the structure, organization, and location of universities [2,3]. Although racial discrimination may still exist among South Africans, the most significant divide among Black South Africans is the difference between the poor and the wealthy, the lower class, and the middle class. This divide results in imbalanced access to opportunities and basic infrastructure [2,6]. Social class influences how we live and experience life in general. For instance, the funding formula of the National Student Financial Aid Scheme (NSFAS) excludes some South African university students based on the social status of their parents. This group of students is commonly referred to as "the missing middle" because of the structuring of the funding policy that makes them neither poor to receive assistance from the government nor wealthy enough to fund themselves [11]. This type of policy-based discrimination is not limited to higher education; it emerges in other policies as well. Reducing discrimination would necessitate a call for the emancipation of the marginalized through the improved provision and access to digital infrastructure, particularly in the service of education, because education was used as a means of control to promote white supremacy at the expense of the non-white population [3]. Access to resources alone may not be sufficient to bridge the divide [6]. Extensive training of lecturers and students [15] would thus be essential to use technology effectively for teaching and learning in the context of RBUs. The critical realist analysis of student participants' empirical data confirms that students who register in RBUs are low-income students who deliberately choose to study in RBUs because they cannot afford to study in other universities [3,4]. This finding coincides with [22]'s assertion that low-income students cannot afford to study at expensive universities. Many students in RBUs in South Africa are thus working-class students [2,3]. The student body is generally diverse [20] in all institutional types. Students possess different attributes such as learning styles, attitudes, perspectives, values, and goals [20]. In addition, students' personal, religious, and cultural values underpin their behaviors. Student participant 1, for instance, asserted: " . . . *as a child born and bred in a Christian family of moral values, I cannot cheat in tests and examinations . . . even if my classmates cheat.*" The extract shows how the student participant drew on family values to abide by the university's academic integrity policy during online summative assessments. It would be inappropriate, though, to assume that all students who come from a Christian background would respond to cheating in the same way. Generally, an individual's habitus can thus be understood to reflect their demographic characteristics as well as cultural and social capital [23]. Students' personal characteristics influence the way they behave, perceive and experience university life and ultimately the way they integrate with the university culture [21,24]. Students whose personal values are aligned with the available university structures [4,24], whether political or religious, are likely to feel more connected to the university compared to students who do not find their associates [21]. The way students perceive and experience integration with the university is directly linked to how they perform in their studies.

During the pandemic, students were forced to study at home. This meant that what would have been conducted at the university had to be performed at home because of the pandemic. This change caused an increased workload for both teachers and students [18,25–27].

Students' home conditions were completely different [20,25]. Many students could not access the internet [9] when they were at home because of the unavailability of a network connection [28] and electricity [15], and sometimes the unaffordability of data [25] after what was offered by the university was depleted. The findings suggest that the early depletion of data could be attributed to the computer illiteracy of the student and the inefficient pedagogical approaches of lecturers. The computer-illiterate student could spend a large portion of the data watching "how to ... " videos on YouTube because they are computer-illiterate. Alternatively, lecturers could use unproductive live lecture methods that require long time attendance to address issues that could possibly be addressed in less time.

The critical realist lens allows us to connect the unaffordability of data to the socioeconomic status of students' homes. Students who come from low-income homes are likely to experience this limitation more than students who come from middle-class homes. Social class can thus be seen to play a role in shaping students' experiences of ERT. Early depletion of data coupled with the unaffordability of data would mean that the student is excluded from the teaching and learning process. In such cases, students use night data to download lecture recordings. The disadvantage of relying on downloaded lecture recordings is that students do not have the opportunity to engage in the discussion. They passively observe what took place during the class and learn from that. Perhaps if they were part of the discussion, they could have experienced the class differently. ERT could be seen to benefit students who can afford to buy data while disadvantaging those who cannot afford to. The subsequent unintended pedagogical consequence of ERT in this case is the lack of adequate epistemological access by low-income students. This implies that the underlying principle of not leaving any student behind was not adequately observed, since middle-class students could be seen to benefit from class attendance more than working-class students. ERT thus intensified the digital and educational divide between low-income and middle-class students. It has been acknowledged that the lower levels of technology exposure among students coupled with the lower financial position of the institution pose substantial impediments to bridging the digital divide [2]. In this article, we argue that critical reflection on the digital divide and attempts to address it should take a contextual rather than a technology-centric perspective. Providing laptops and internet access alone is insufficient to bridge the digital divide [29]. This provision is essential, but it should be accompanied by skill development, a shift in mentality, and an acknowledgment of the magnitude of the problem [2].

Working-class students who stay on campus are likely to have more chances of accessing resources, such as the library and computer labs, compared to students who stay off campus [30]. In addition, students who stay on campus are more likely to know senior students who studied the same courses, and subsequently have better chances of peer support and integration into the university culture [21]. Moreover, they are likely to be involved in extramural activities, in so doing expanding their social network. Students are social beings [4,31], so the sense of belonging is critical to their well-being. During the pandemic, students returned to the university campuses only after the lockdown restrictions were relaxed. During the hard lockdown, social media played a major role in linking peers from different geographical areas and the availability of data and network were crucial.

ERT was thus rated lower than traditional lectures in relation to students' engagement in class activities [9,22]. Students who study at home report less positive university experiences, lower levels of engagement in academic studies, student social life, and extracurricular activities, and fewer opportunities to develop social and cultural capital and learning through informal interaction [20,23]. Academic and social integration during the pandemic was essential to determine whether students continued pursuing their goals in the university or gave up the academic years [21]. Restrictions on gathering and traveling prevented physical collaboration between students, lecturers, and research and conference attendance resulting in social loneliness [32]. This resulted in students and lecturers feeling alienated and suffering from mental health issues, such as depression and anxiety, arising from increased stress [26], workloads [32], and isolation [28]. Ultimately, lecturers took



sick leave and students ran the risk of dropping out [26]. It is for this reason that the university's counseling facilities were critical to assist students and academics emotionally. However, some students, especially first-year students, were not aware of the existence of such facilities, while others preferred not to use them because of the stigma associated with them. In many respects, COVID-19 exacerbated inequality [7,26] in varying levels of family support for students during the pandemic [33]. Again, the critical realist lens allows us to associate the differentiated family support of students with their family's socioeconomic standing where middle-class families were seen to support students more significantly than working-class families.

Different lecturers' pedagogical approaches influenced the way students experienced ERT [7,10,30,31,34]. Lecturers began to use media or teaching methods that they were familiar with and perceived as useful and appropriate [32,34]. Some lecturers had no idea how to transform their existing teaching resources into online learning spaces [5,8,9]. Subsequently, such lecturers taught in online classes in the same way they would teach in traditional lectures [26]. The lecturer would spend almost ninety minutes of teaching trying to engage students in discussions that most of the time were not successful because students could not participate in them. This pedagogical approach consumed a lot of data. On the contrary, many students expressed dissatisfaction with the lack of engaging, collaborative, and interactive class discussions [22] in the context of UBUs.

The results of the study have explained some of the reasons for students not participating in live lectures. One of the reasons was a lack of confidence in the language of instruction. The opposite is true in UBUs. Most students showed their willingness to participate in class discussions. Prior schooling experiences emerge as a causal mechanism for students to experience live classes differently. Universities are not the same and, as a result, strategies that are supportive at one university might not be appropriate for another. One university might be privileged and have easy access to technology and motivated lecturers; other universities might have to address students that were barely reachable since they did not have proper means to access the internet while staying at home [32].

Regarding assessments, some students stated that online assessments were much easier than traditional venue-based assessments. This finding is also directly linked to the pedagogical expertise of lecturers. Some lecturers found it challenging to assess students authentically online [28]. For instance, while they could be aware of assessment practices such as open-book examinations [26], they might not have been equipped to set questions for that kind of assessment. The underlying principle of setting open-book assessments is that students should not be able to find direct answers online if good questions are asked. Lecturers need training [15] in designing and developing good assessment questions. Alternatively, some lecturers would use discussion forums to minimize the amount of data consumed during live sessions. Again, only a few students participated in online discussion forums.

The overall finding thus was that some lecturers lacked pedagogical knowledge and experience in teaching online [26,28]. This implies that pedagogical training is essential [26] if lecturers must teach and assess effectively in online environments. Lack of adequate engagement in learning and limited authentic assessment practices that encourage deep approaches to learning would mean that epistemological access to students' learning is questionable during the era of ERT.

The findings and discussion provided above show how the personal attributes of students and lecturers enabled and constrained their academic and social integration into the university during the pandemic [20]. Lecturers who used blended learning before the pandemic transitioned to ERT differently than lecturers who used blended learning for the first time [30] during the pandemic. In addition, the way lecturers managed their classes could have been experienced differently by students depending on students' prior experiences [20] of blended learning. For instance, the technologically experienced lecturer reported using discussion forums to engage students and tried innovative ways to minimize data consumption, whereas the less technologically competent lecturer was not so effective in engaging students and saving data.

We could see through the critical realist lens that the socioeconomic conditions of students emerged as the conditioning structures for the experiences and observations that emerged for both students and lecturers. For instance, poor attendance of virtual classes by students was seen to have been triggered by infrastructural and socioeconomic constraints, such as the unavailability of network connection, the unavailability of electricity [31], and the unavailability of data [26]. The cultural constraint associated with the social construction of gender roles in students' homes, such as looking after cattle by male students and doing household chores [13,30] by female students, also surfaced in the study. The discrimination of females compared to their male counterparts in many aspects of social life is prominent in African countries and is well-recorded in the literature. Therefore, the higher education systems should not by any means perpetuate past inequalities. These technological and social structures are enduring and are likely to constrain future adoption of remote teaching beyond the pandemic if students continue to study from home. The critical analysis lens has allowed us to go beyond observations and experiences reported by student and lecturer participants to understand the underlying structures from which the events emerged. For instance, the recording of live online sessions was reported to have caused some students to stop participating in discussions. This might be because they were not confident in speaking the language of instruction or because they were not given the freedom to speak in their home language in cases where both lecturers and students understood students' home language. Lack of confidence to speak in public could be attributed to students' prior schooling and personal attributes that were discussed earlier in the article. Such events, observations, and experiences are linked to the structures at the level of the real, such as the family's socioeconomic conditions.

The findings of this study resonate with the findings recorded on Chinese middle school students. Rural students reported lower levels of achieving learning outcomes in e-learning courses than their urban peers. Although the study was not conducted in a university setting, it confirms the existence of the digital outcome divide between rural and urban students. Universities recruit their students from these schools. This confirms the correlation between high schools and universities. The primary causes of the digital outcome divide are differences between rural and urban students in habitus, (i.e., intrinsic motivation), forms of capital, including cultural, (i.e., e-learning self-efficacy), and social capital, (i.e., parental and teacher support) [20]. Thirdly, it was confirmed that these structures could be interpreted as the causal mechanisms for the digital outcome divide between urban and rural students, and that e-learning self-efficacy, intrinsic motivation, and parental support were the most influential structures in the rural-urban digital achievement gap in the e-learning context [22]. The digital outcome divide is understood as the difference in achieving learning outcomes because of the influence of other forms of the social divide [22]. The working-class students missed out on opportunities to engage meaningfully with learning materials because, for a variety of reasons, they were unable to attend all live sessions [35]. As a result, their epistemological access may be rated lower than their counterparts [19]. Inadequate assessment practices in the online environment by some lecturers, though, may fall short of identifying this gap in epistemological access. Additionally, the digital divide [3,29,36,37] that existed prior to the pandemic was exacerbated by the shift to ERT [25]. While some working-class students used their bursary funds to support their families, middle-class students purchased more advanced educational technologies. Furthermore, insufficient lecturer training [15] could have resulted in lower quality standards of teaching and assessment than could have been possible in traditional face-to-face classes. Another troubling finding was that the legitimacy of online assessments could not be guaranteed because assessments were not monitored [38]. It might be possible that students in some courses assisted one another in completing online summative assessments. The availability of proctoring software does not eliminate academic dishonesty completely. Recent research conducted in South Africa [38] confirmed that several factors contributed to academic dishonesty among students during the pandemic. Among these factors was (1) the availability of online content with ease. (2) Students felt overloaded and anxious.

(3) Lack of invigilation. (4) Ineffective time management. (5) Lecturers recycle questions and allocate excessive time for assessments. (6) The academic inexperience of students. (7) Having difficulty with technology [38]. These are some of the unintended pedagogical consequences that participants' interviews revealed. Future research should be designed and developed to address these issues. The study confirms that lecturers require more training [15] not only to be technologically competent, but also to be pedagogically competent in the online environment to manage these challenges. Furthermore, universities, especially RBUs, should have plans in place to accommodate students who are unable to engage with online materials due to home circumstances, as discussed in the study.

## 6. Limitations

ERT brought many opportunities for students and lecturers in all types of higher education institutions, including RBUs. However, they are not discussed in this study; they were excluded from the purpose of the study, which was to highlight the unintended pedagogical consequences of ERT. The unintended pedagogical consequences discussed in this article are neither exhaustive nor exclusive to the institution under study; they are experienced globally, including at historically privileged institutions. For instance, a study conducted at a historically advantaged university in South Africa confirmed that most students who participated in the study experienced more disadvantages than benefits. A total of 2744 complaints ranged from distractions to inequitable living and working environments and a total of 1584 benefits ranged from adaptability to self-directed learning [22]. Our study was significant to contribute to the dearth of knowledge produced by rural universities in South Africa [14]. Its qualitative nature prevents the generalization of its findings. Its findings are only applicable to comparable situations. Another limitation is the employed framework. It did not elaborate on the participants' technological experiences and psychological states during the transition. Future research could be conducted to gain a comprehensive understanding of these concepts.

## 7. Conclusions

The digital divide in South Africa is diverse, requiring the liberation of marginalized communities through improved access to digital infrastructure, primarily in basic and higher education [2]. To use technology effectively for teaching and learning at RBUs, lecturers and students would, therefore, require extensive training [15]. Students coming from low-income homes should not be perceived as defective stakeholders that need to be fixed. The social and cultural structures beyond their control support middle-class students and marginalize students from low-income homes. Higher education policies and practices should not discriminate against any students; instead, they should be designed to emancipate them. Universities, academics, and support staff were "underprepared" to perform their duties effectively during the pandemic. The challenges RBUs experience are influenced largely by the higher education policies that require drastic restructuring to emancipate rural communities. The key findings drawn from participants' interview data presented above are that (1) Students' and lecturers' home conditions inclined the way students and lecturers perceived, practiced, and experienced ERT, (2) students' and lecturers' individual attributes influenced how students and lecturers perceived, practiced, and experienced ERT, (3) students' and lecturers' blended learning experiences before the pandemic determined the way lecturers perceived, practiced, and experienced ERT, (4) the training and support that the university provided to students and lecturers were connected to the way students and lecturers perceived, practiced, and experienced ERT, and (5) the teaching, learning, assessment practices, and policies of the university affected students' and lecturers' perceptions, practices, and experiences of ERT.

Although ERT was meant to save the academic years during the pandemic and accommodate every student, the way it was implemented may not have been completely productive in RBUs. As a result, it constructed unintended pedagogical consequences, such as, (1) the exclusion of low-income students in the process of active teaching and learning,

(2) equipping middle-class students with better chances of success than working-class students, (3) distressing female students and lecturers more than their male counterparts, and (4) unproductive assessment practices that may have fallen short to assess students' learning comprehensively.

## 8. Recommendations

To confront these challenges, (1) university policies should concentrate on responding to the concerns of the digital divide [22,36], (2) lecturers should be given more autonomy to be innovative in their teaching and assessment practices, (3) flexible methods of assessment should be encouraged, (4) less technological pedagogical models that may be better suited to areas lacking a reliable internet connection should be explored because technology is not always a viable solution in all contexts, (5) infrastructure that supports hybrid blended learning should be acquired to accommodate students who prefer to learn online and through on-contact lectures simultaneously, (6) most importantly, instructional designers who train lecturers should be well conversant in the latest technologies and pedagogical approaches that are appropriate for the teaching and learning environments of the university. Ultimately, lecturers should be well-equipped both pedagogically and technologically, and (7) hybrid modes of teaching and assessment should be considered to take advantage of the best practices of online and contact education, blending them seamlessly to accommodate both students with and students without the necessary resources to engage meaningfully online.

**Author Contributions:** Conceptualization, S.T.P. and S.R. emerging from Ph.D. study; methodology, S.T.P.; data production, S.T.P.; validation, S.T.P.; formal analysis, S.T.P. and S.R.; data curation, S.T.P.; writing—original draft preparation, S.T.P. and S.R.; writing—review and editing, S.T.P. and S.R.; visualization, S.T.P.; supervision, S.R.; project administration, S.T.P. All authors have read and agreed to the published version of the manuscript.

**Funding:** This research received no external funding.

**Institutional Review Board Statement:** The study was conducted in accordance with the approval of the Ethics Committee of the university where the first author is registered as a Ph.D. student.

**Informed Consent Statement:** Informed consent was obtained from all subjects involved in the study.

**Data Availability Statement:** The data for this article emerges from the larger Ph.D. study and are protected by the ethical clearance certificate by the institution where the Ph.D. is registered.

**Acknowledgments:** The first author of this article would like to acknowledge the Learning and Teaching Directorate at the university where he is employed for funding writing retreats in support of novice researchers. This article is the product of such interventions by the university.

**Conflicts of Interest:** The authors declare no conflict of interest in this study.

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
