# Peer review of "Unintended Pedagogical Consequences of Emergency Remote Teaching at a Rural-Based University in South Africa"

_education, doi:10.3390/educsci12110830_

Round 1
Reviewer 1 Report
This is a case study, good to improve:
1.The research context of ERT
2. The description of research group (students and teachers, years, studied, experienced s.a). However is a small research group, so the results are not relevant to be generalized.
3. The research instrument is not described in relation to the research group
3. What are the research questions? The research scopes?
4. It is necessary to improve the references according to the results.
Paniagua, A. and D. Istance (2018), Teachers as Designers of Learning Environments: The Importance of Innovative Pedagogies, Educational Research and Innovation, OECD Publishing, Paris, https://doi.org/10.1787/9789264085374-en
Tripon, C. Supporting Future Teachers to Promote Computational Thinking Skills in Teaching STEM—A Case Study. Sustainability 2022, 14, 12663. https://doi.org/10.3390/su141912663
Author Response
Thank you for your critical review of our article. See the attached responses.

Reviewer 2 Report
This article is very clear in its proposal and structure. The methods and analysis area adequate, and the articulation of critical realism with student integration theory is sound. The main issue here is that many of the findings regarding the difficulties with emergency remote teaching of rural universities are also valid for those students and teachers engaging in EMR in urban universities, even in European countries. Hence, the paper should highlight a) the differences between urban/rural based universities, not only in terms of student recrutement base and backgound resources; b) differences between rural background and middle class background students in South Africa, that are invoked relatively with resource to other literature but that are not documented with studies from this specific reality. My suggestion is to look for literature that studies EMR with more privileged student samples, and stress the differences (since for some students, EMR was atually better and they have adapted to it, not only due to better digital literacy and technological resources, but also for convenience). This refocus will make the paper more relevant to international audiences.
I also consider that the paper should make some recommendations regarding specific pedagogies, both presencial and at distance to be adopted with rural based students, since disasters and catasthophes are increasingly common and education in emergencies is a distinct field with its own specificities.
Author Response

(The authors gave the same response as above.)

Round 2
Reviewer 1 Report
1.The research context of ERT- poor references
2. The research instrument it is necessary to be more detailed (questions, scopes)
3. It is necessary to improve the references, there are a small number and not all are relevant according to the results. Examples:
Tripon, C. Supporting Future Teachers to Promote Computational Thinking Skills in Teaching STEM—A Case Study. Sustainability 2022, 14, 12663. https://doi.org/10.3390/su141912663
Paniagua, A, Istance, D. (2018), Teachers as Designers of Learning Environments: The Importance of Innovative Pedagogies, Educational Research and Innovation, OECD Publishing, Paris, https://doi.org/10.1787/9789264085374-en
Author Response
Thank you for your review of the article. Your comments are very useful.
Find our responses attached.
